# Experience and Prevalence of Dental Caries in 6 to 12-Year-Old School Children in an Agricultural Community: A Cross-Sectional Study

**DOI:** 10.3390/children8020099

**Published:** 2021-02-03

**Authors:** Juan José Villalobos-Rodelo, Martha Mendoza-Rodríguez, Rosalina Islas-Zarazúa, Sonia Márquez-Rodríguez, Mariana Mora-Acosta, América Patricia Pontigo-Loyola, María de Lourdes Márquez-Corona, Carlo Eduardo Medina-Solís, Gerardo Maupomé

**Affiliations:** 1School of Dentistry, Autonomous University of Sinaloa, Culiacan 80040, Mexico; villarodelo@yahoo.com.mx; 2Department of Epidemiology, Institute of Social Security and Services for Government Workers, Culiacan 80000, Mexico; 3Academic Area of Dentistry, Health Sciences Institute, Autonomous University of Hidalgo State, Pachuca 42160, Mexico; riz751@hotmail.com (R.I.-Z.); cdsoniamr@hotmail.com (S.M.-R.); mtramarianamoraa@hotmail.com (M.M.-A.); americap@uaeh.edu.mx (A.P.P.-L.); lmarquez@uaeh.edu.mx (M.d.L.M.-C.); 4Advanced Studies and Research Center in Dentistry “Dr. Keisaburo Miyata”, School of Dentistry, Autonomous University of State of Mexico, Toluca 50000, Mexico; 5Richard M. Fairbanks School of Public Health, Indiana University/Purdue University, Indianapolis, IN 46202, USA; gmaupome@iu.edu; 6Indiana University Network Science Institute, Bloomington, IN 47408, USA

**Keywords:** oral health, dental caries, schoolchildren, agricultural workers, Mexico

## Abstract

Objective: To describe the experience and prevalence of dental caries in schoolchildren aged 6–12 years belonging to agricultural manual worker households. Material and Methods: A comparative cross-sectional study was conducted in two groups of schoolchildren: One considered “children of agricultural worker migrant parents” (*n* = 157) and the other “children of agricultural worker non-migrant parents” (*n* = 164). Epidemiological indices for dental caries were calculated for primary (dmft) and permanent (DMFT) dentitions, and compared in terms of age, sex, and the Simplified Oral Hygiene Index (SOHI). Two binary logistic regression models for caries prevalence in primary and permanent dentitions were generated in Stata. Results: For primary dentition, we observed the following dmft index: Non-migrants = 1.73 ± 2.18 vs. migrants = 1.68 ± 2.14. Additionally, we recorded the following caries prevalence: Non-migrants = 59.1% vs. migrants = 51.3%. For permanent dentition, we observed the following DMFT index: Non-migrants = 0.32 ± 0.81 vs. migrants = 0.29 ± 0.95. Further, we recorded the following caries prevalence: Non-migrants = 17.6% vs. migrants = 12.8%. No differences were observed for either dentition (*p* > 0.05) in caries indices and their components or in caries prevalence. When both caries indices (dmft and DMFT) were combined, the non-migrant group had a higher level of caries experience than the migrant group (*p* < 0.05). No relationship (*p* > 0.05) with migrant status was observed in either multivariate models of caries prevalence. However, age did exhibit an association (*p* < 0.05) with caries. Only the plaque component of SOHI was associated (*p* < 0.05) with caries in permanent dentition. Conclusions: Although over half of school children from agricultural manual worker households had caries in either or both dentitions and a considerable proportion were untreated lesions, the prevalence levels were somewhat lower than other reports from Mexico in similar age groups. No statistically significant differences were found in caries experience or prevalence in either dentition between non-migrant and migrant groups.

## 1. Introduction

The oral health of the world’s population has not improved in recent decades, despite certain nuances in its distribution, and oral conditions continue to be a major challenge for health systems. The cumulative burden of oral conditions increased dramatically between 1990 and 2015: The number of people with untreated oral conditions increased from 2.5 billion in 1990 to 3.5 billion in 2015 [1]. Dental caries has both a high prevalence and incidence, mainly in those with a socioeconomic disadvantage, and continues to be a serious global health problem [2]. In Latin America [3,4,5] and Mexico, school-age children and adolescents have considerable treatment needs and low experiences of dental services for restorative and preventive treatment [6,7]. Caries impacts diverse aspects of life; for example, dental caries may be associated with dental pain, failure to thrive, and negative effects on body weight and height, and may affect one’s quality of life [8,9].

Due to the lack of employment opportunities, wage inequalities, lack of access to services, and higher demand for workforces in areas with a higher level of development, people often need to migrate [10]. Within internal migration (that is, within the same country), rural–rural migration linked to the agricultural labor market has a special relevance [11]. In Mexico, it is estimated that there are more than 5.9 million people in this situation; almost half are minors [12]. Mexican migrant agricultural workers are one of the poorest and most marginalized social groups in the country: Living conditions and health risks are precarious, and they lack secure access to public health services. Such limitations translate into a greater risk for poverty, nutritional deficiencies, and diseases [13]. These conditions are similar to situations in other parts of the world for this type of worker [14].

Such heightened health vulnerability among agricultural workers in Mexico is derived, first and foremost, from marginalization and poor living conditions prevailing in their communities of origin; they usually move out of rural areas, geographically isolated, in the poorest states of Mexico. Social deprivation is greater in this migrant agricultural worker population compared to the general population: 72.4% are affected by at least one social deprivation factor (e.g., insufficient food; limited educational opportunities; restricted access to health services; reduced quality and space of the household; and limited or only basic utility services of the household, such as running water or piped sewage) [15]. Another 22.1% display three or more such factors. Given these precarious circumstances, rural–rural migration, conducted to gain opportunities in the agricultural labor market, represents one of the few opportunities to secure a job (often temporary) in order to support the family. However, job opportunities derived from migration are often fraught with the instability of the rural labor market [16].

In research outside Mexico, a worse oral health status has been found in children of agricultural migrants compared to other population groups. For example, children of migrant farm workers aged 5 to 14 had more decayed teeth and fewer restored teeth than American schoolchildren overall [17]. Moreover, other studies have found worse dental caries conditions in children residing in rural regions than in urban areas [18,19]. Oral health and access to dental care among migrant agricultural workers in other countries are characterized by marked differences with other population groups [17,20,21,22], often having the greatest unmet health needs [23]. In Mexico, there are sparse or no reports about the oral health status of these workers. The present study aimed to determine the experience and prevalence of dental caries in schoolchildren from agricultural worker households between 6 and 12 years of age, and ascertain the influence of migrant agricultural and non-migrant agricultural statuses in a community of Sinaloa, Mexico.

## 2. Materials and Methods

### 2.1. Design and Study Population

A cross-sectional study was conducted in schoolchildren aged 6–12 years. Data collection took place from February to May 2018. The community had a relatively homogeneous socioeconomic level. The sampling frame included the six public schools from the Villa Benito Juárez community in Navolato, Sinaloa, Mexico, with a population of 2066 students. Children were students aged 6 to 12 enrolled in such schools. Exclusion criteria were having fixed orthodontic appliances, or syndromes or craniofacial malformations with an impact on oral diseases. “Migrant children” were children of agricultural workers, born outside the state of Sinaloa, for whom both parents were from any state in Mexico other than Sinaloa, and who had lived in the state for no more than 5 years. Those children belonged to households originating in areas where there is higher unemployment and a lack of services, generally from mountainous areas with difficult physical access; many of them belong to an indigenous ethnic group. For “native children”, criteria were that both parents were born in Sinaloa, with long-standing residence. We refer to the latter as “native”. The different statuses of migrant and non-migrant were ascertained through a survey offered to parents of school children. A total of 767 native children and 189 migrants were identified in the study.

The sample size was calculated based on an estimation of the difference in proportions, considering a proportion (p1) of 0.54 in native families and p2 of 0.35 in migrant families (proportions estimated in a pilot study), a 95% confidence level, and a power of 90%. An expected loss ratio of 15% was considered. The estimated sample size in each group was 167. After applying the inclusion and exclusion criteria, *n* = 157 “children of agricultural migrant parents” and *n* = 164 “children of non-migrant parents” were studied. A questionnaire was administered to parents/guardians with prior authorization from school authorities and parents’/guardians’ signed informed consent.

### 2.2. Variables and Data Collection

The clinical examination was performed by four dentists using a flat dental mirror and a dental probe, with natural light. They were trained and standardized in terms of the criteria used (inter- and intra-observer concordances, with a kappa value between 0.81 and 0.91).

We followed standard guidelines for assessing the oral health status in a population, using the WHO Oral Health Surveys Basic Methods manual [24]. Epidemiological indices recommended by WHO for dental caries in primary dentition (decayed teeth, extracted/indicated for extraction and filled, or the primary dentition (dmft) index) and permanent dentition (decayed, missing and filled teeth, or the permanent dentition (DMFT) index) were calculated. The dependent variables were the caries experience (mean dmft or DMFT) and prevalence (dmft > 0 or DMFT > 0) in both dentitions. In the clinical evaluation, the presence of plaque and dental calculus was measured using the Simplified Oral Hygiene Index (SOHI) [25]. In children with primary and mixed dentition, the modified SOHI was used [26]. The independent variables were the migrant status (0 = non-migrant and 1 = migrant), age (6 to 12 years), and sex (0 = boys and 1 = girls).

### 2.3. Statistical Analysis

For a descriptive analysis of variables, measures of frequency, central tendency, and dispersion were used, depending on the scale of measurement of variables. Chi-square, Mann–Whitney, Kruskall–Wallis, and non-parametric trend tests were used for bivariate analyses. To generate multivariate models for the caries prevalence of primary and permanent dentitions, binary logistic regression models were used, where odds ratios with their 95% confidence intervals were calculated. The fit of the models was evaluated with the Hosmer and Lemeshow goodness of fit test [27]. Analyses were performed with Stata 14 software.

### 2.4. Ethical Considerations

This study complied with specifications provided in the general health law on research in Mexico, and with the Helsinki scientific principles. The protocol was approved by the Institutional Review Board, School of Dentistry at the Autonomous University of Sinaloa. Written consent was obtained from parents/guardians of children.

## 3. Results

Three hundred and twenty-one schoolchildren aged 6 to 12 years were included in the study; the mean age was 8.98 ± 1.95 and 53% were girls. The study population included 164 children from non-migrant agricultural worker households and 157 from migrant households. No difference was observed in the distribution by age and sex between groups (Table 1).

Table 2 shows the distribution of caries indices and their components. In the primary dentition, the average dmft index for the total study sample was 1.71 ± 2.16; for the non-migrant group, dmft was 1.73 ± 2.18; and for the migrant group, dmft was 1.68 ± 2.14. In terms of the permanent dentition, the mean DMFT index for the total study sample was 0.30 ± 0.88; for the non-migrant group, it was 0.32 ± 0.81; and for the migrant group, it was 0.29 ± 0.95. No statistically significant differences were observed (*p* > 0.05) in the total indices or in their components between the groups in either dentition. When dmft and DMFT were combined, non-migrant children had a higher level of mean caries experience than migrant children.

The dmft and DMFT indices (caries experience) organized by age and sex are presented in Table 3. In the non-parametric test for trends, it was observed that the mean dmft decreased as the age increased, both in the non-migrant group (z = −2.90, *p* = 0.004) and in the migrant group (z = −2.20, *p* = 0.028); the mean DMFT increased with age in the non-migrant group (z = 3.04, *p* = 0.002), but not in the migrant group (z = 1.31, *p* = 0.189). Using the Kruskal–Wallis test, we found a significant difference in the dmft distribution across age in the non-migrant group (*p* = 0.0115), but not in the migrant group (*p* = 0.2556). Considering the DMFT distribution across age, no significant differences were found in either the non-migrant (*p* = 0.0528) or migrant groups (*p* = 0.6618). When we contrasted the caries experience in both primary and permanent dentition between non-migrant and migrant groups (dmft 1.73 ± 2.18 vs. 1.68 ± 2.14, *p* = 0.6400; DMFT 0.32 ± 0.81 vs. 0.29 ± 0.95, *p* = 0.2555), no significant differences were identified. We can see in Table 3 that there were no significant differences in the distribution of indices for primary or permanent dentition across sex (*p* > 0.05).

In Table 4, we present analyses of the caries prevalence in both dentitions (dmft > 0 and DMFT > 0). The caries prevalence in primary dentition was 59.1% for non-migrants and 51.3% for migrants (*p* = 0.211). For the permanent dentition, the caries prevalence was 17.6% for non-migrants and 12.8% for migrants (*p* = 0.227). In the non-migrant group, the caries prevalence decreased with age for the primary dentition (non-parametric test for trends, z = −2.17, *p* = 0.030) and increased with age for permanent teeth (non-parametric test for trends, z = 2.94, *p* = 0.003). In the migrant group, no differences (*p* > 0.05) were observed by age in either dentition. There was no difference in the caries prevalence across sex (Table 4).

Table 5 presents a crude and adjusted logistic regression analysis for the caries prevalence in both dentitions. In primary dentition, it was observed that, when the age increased, the odds of presenting caries decreased (Odds Ratio (OR) = 0.78, 95% Confidence Intervals (CI) = 0.66–0.91); no other variable was significant in the final model. In permanent dentition, for each year of age, the odds of presenting caries increased 1.29 times (95% CI = 1.09–1.54); in addition, for each unit of increase in the SOHI plaque component, the likelihood of presenting caries increased 2.66 (95% CI = 1.01–7.02) times. The migrant condition had no effect on the caries prevalence in either of the two models.

## 4. Discussion

While the caries prevalence was over 50% and a considerable proportion were untreated lesions in our study, the figures were somewhat lower than in other reports from Mexico on similar age groups. Additionally, we did not observe significant differences across the non-migrant and migrant groups. Studies on oral health in this type of population are scarce, so comparisons of our results with prior literature are limited. The caries levels appear to be similar to studies previously conducted in Mexico and other countries. For example, another study from Mexico focused on a general child population of a similar age reported a greater experience and prevalence of dental caries in primary than permanent dentition [6], with a range of the dmft index between 0.73 and 5.35 at 6 years of age, and DMFT index between 0.52 and 3.67 at 12 years of age. Other reports have suggested that the caries prevalence in permanent dentition ranges from 70% to 85% among 12-year-old children [28]. Based on this contrast, the caries indices and prevalence found for primary dentition are within the ranges observed in other studies, but not for permanent dentition. The latter were lower than other reports from the general Mexican population. In another Mexican indigenous minority population, high caries rates and treatment needs were observed [29]. Other studies have mentioned that migrant children visit the dentist less frequently and when they are older, they are less likely to have dental insurance, but have a higher incidence of decayed teeth and both a disproportionate prevalence of decayed teeth and unmet dental need, in comparison to children from ‘native’ households [30].

The Americas continue to be one of the most inequitable regions of the world, with millions not having adequate access to health services. Such barriers in access to care preferentially affect poorer population groups [31]. This differential scenario can be distinguished, even across states in Mexico [32]. Our basic hypothesis linking the migration status with a greater caries experience and prevalence was not confirmed. On the one hand, this may be due to the moderation of influencing factors, such as biological or social aspects, lifestyles, practices, behaviors, and access to health services. It is possible that differences in sugar intake and the availability of fluoride (from toothpastes, drinking water, and foods) between the two study populations could also be at play. Additionally, a factor likely to have an influence is a program aimed at migrant agricultural workers and their families to prevent and treat various diseases. This is a Federal Government strategy implemented through the Mexican Institute of Social Security; it does include limited dental care services. On the other hand, migrant and non-migrant children may share characteristics afforded by residing in the same community, thereby offsetting caries-relevant differences that may have existed for households of migrant children in their communities of origin. Limited data on health service coverage suggest unequal improvement in various Latin American countries (including Mexico), with some impact on health and social inequalities [31], possibly improving oral health trends.

Despite accruing valuable caries data about an otherwise rarely studied population group, the present research has some limitations. One of the limitations of this study was its cross-sectional design, which did not allow for cause–effect relationships to be established between caries and a migrant status. We were unable to gather reliable information from study participants to take into account variables such as tooth brushing with fluoride toothpaste, dietary patterns, access to services, and actual use, and whether children in the government support program received specific dental services and their frequency. In terms of the study’s strengths, we examined a group that is often left out of epidemiological studies in Mexico, even when such studies incorporate low-income Mexican population groups. Future studies could incorporate larger sampling frameworks, non-clinical variables, and more diverse comparison groups.

## 5. Conclusions

In conclusion, although over half of school children from agricultural manual worker households had caries in either or both dentitions and a considerable proportion were untreated lesions, the prevalence levels were somewhat lower than in other reports from Mexico on similar age groups. No statistically significant differences between non-migrant and migrant groups were found in terms of the caries experience and prevalence in either dentition. Because dental caries remains a pervasive problem for both groups, the need to implement school and non-school health programs is apparent, ideally addressing both preventive and curative aspects.

## Figures and Tables

**Table 1 children-08-00099-t001:** Age and sex distribution of schoolchildren.

Variable	Non-Migrant (*n* = 164)	Migrant (*n* = 157)	All	*p* Value
Age	8.79 ± 1.87	9.18 ± 2.03	8.98 ± 1.95	0.0965 *
Sex				
Boys	75 (49.7)	76 (50.3)	151 (47.9)	
Girls	89 (52.3)	81 (47.7)	170 (53.0)	*p* = 0.631 ^†^

* Mann–Whitney test. ^†^ Chi square test.

**Table 2 children-08-00099-t002:** Distribution of caries indices (primary dentition (dmft) and permanent dentition (DMFT)) and their components among non-migrant and migrant schoolchildren.

Variable	Non-Migrant	Migrant	All	*p* Value *
dmft index	1.73 ± 2.18	1.68 ± 2.14	1.71 ± 2.16	>0.05
Decayed primary teeth	0.88 ± 1.60	0.74 ± 1.17	0.81 ± 1.43	>0.05
Missing primary teeth	0.04 ± 0.20	0.06 ± 0.30	0.05 ± 0.25	>0.05
Primary teeth indicated for extraction	0.44 ± 0.87	0.47 ± 0.93	0.45 ± 0.89	>0.05
Filled primary teeth	0.38 ± 0.98	0.40 ± 1.12	0.39 ± 1.04	>0.05
DMFT index	0.32 ± 0.81	0.29 ± 0.95	0.30 ± 0.88	>0.05
Decayed permanent teeth	0.15 ± 0.53	0.18 ± 0.67	0.16 ± 0.60	>0.05
Missing permanent teeth	0.01 ± 0.15	0.00 ± 0.00	0.00 ± 0.11	>0.05
Filled permanent teeth	0.15 ± 0.53	0.10 ± 0.52	0.13 ± 0.53	>0.05
dmft + DMFT index	1.82 ± 2.26	1.50 ± 2.24	1.66 ± 2.26	0.0381

* Mann–Whitney test.

**Table 3 children-08-00099-t003:** Caries experience in primary and permanent teeth by age group and sex in the migrant and non-migrant groups.

	Mean dmft			Mean DMFT		
Age	Non-Migrant	Migrant	*p* Value	Non-Migrant	Migrant	*p* Value
6	2.80 ± 2.82	1.42 ± 1.60	Non-migrant *	0.15 ± 0.48	0.07 ± 0.27	Non-migrant *
7	1.53 ± 1.73	2.21 ± 2.21	age vs. dmft	0.07 ± 0.37	0.10 ± 0.31	age vs. DMFT
8	2.24 ± 2.61	2.12 ± 2.72	*p* = 0.0115	0.27 ± 0.64	0.15 ± 0.61	*p* = 0.0528
9	1.51 ± 2.00	1.63 ± 1.92		0.23 ± 0.81	0.23 ± 0.88	
10	1.88 ± 1.93	1.54 ± 2.38	Migrant *	0.36 ± 0.68	0.50 ± 1.16	Migrant *
11	0.23 ± 0.59	0.36 ± 0.67	age vs. dmft	0.61 ± 1.20	0.60 ± 1.49	age vs. DMFT
12	0.71 ± 0.95	0.6 ± 0.89	*p* = 0.2556	0.76 ± 1.20	0.38 ± 1.17	*p* = 0.6618
Total	1.73 ± 2.18 ^†^	1.68 ± 2.14	*p* = 0.6400 ^†^	0.32 ± 0.81	0.29 ± 0.95	*p* = 0.2555 ^†^
Sex ^†^						
Boys	1.89 ± 2.50	1.81 ± 2.20		0.26 ± 0.74	0.27 ± 0.80	
Girls	1.58 ± 1.83	1.51 ± 2.08		0.37 ± 0.87	0.31 ± 1.08	
	*p* = 0.9966 ^†^	0.3829 ^†^		0.3619 ^†^	0.5974 ^†^	

* Kruskal–Wallis test. ^†^ Mann–Whitney test.

**Table 4 children-08-00099-t004:** Caries prevalence in primary and permanent teeth by age group and sex in the migrant and non-migrant groups.

	dmft > 0		*p* Value	DMFT > 0		*p* Value
	Non-Migrant	Migrant		Non-Migrant	Migrant	
Age						
6	75.0	57.1		10.0	7.6	
7	57.1	64.2	Non-migrant *	3.5	10.7	Non-migrant *
8	62.1	48.0	z = −2.17, *p* = 0.030	20.6	7.6	z = 2.94, *p* = 0.003
9	62.9	47.3		10.0	9.5	
10	72.2	54.5	Migrant *	26.3	21.4	Migrant *
11	15.3	27.2	z = −1.74, *p* = 0.081	28.5	21.7	z = 1.17, *p* = 0.241
12	42.8	40.0		35.2	12.9	
Total	59.1	51.3	X^2^ = 1.5624, *p* = 0.211	17.6	12.8	X^2^ = 1.4577, *p* = 0.227
Mann-Whitney test	*p* = 0.0502	*p* = 0.0825		0.0039	0.2711	
Sex						
Boys	55.0	55.7		14.6	14.4	
Girls	63.0	46.1		20.2	11.2	
Chi square test	*p* = 0.336	*p* = 0.310		*p* = 0.353	*p* = 0.547	

* Non-parametric test for trends.

**Table 5 children-08-00099-t005:** Logistic regression analysis for the prevalence of caries in primary and permanent dentitions.

	Model 1: Caries Prevalence in Primary Dentition			
	OR Crude (95% CI)	*p* Value	OR Adjusted (95%CI)	*p* Value
Migrant				
No	1 *		1 *	
Yes	0.72 (0.44–1.19)	0.212	0.77 (0.45–1.29)	0.327
Age	0.81 (0.69–0.94)	0.007	0.78 (0.66–0.91)	0.002
Sex				
Boys	1 *		1 *	
Girls	1.02 (0.62–1.68)	0.921	1.16 (0.69–1.95)	0.564
SOHI (plaque)	1.84 (0.86–3.94)	0.113	1.82 (0.83–3.99)	0.132
SOHI (calculus)	3.07 (.60–15.70)	0.177	3.82 (0.70–20.65)	0.119
	Goodness of fit test Hosmer–Lemeshow chi2(8) = 10.96, *p* = 0.2042			
	**Model 2: Caries Prevalence in Permanent Dentition**			
Migrant				
No	1 *		1 *	
Yes	0.68 (0.36–1.26)	0.229	0.64 (0.33–1.22)	0.176
Age	1.25 (1.06–1.47)	0.006	1.29 (1.09–1.54)	0.003
Sex				
Boys	1 *		1 *	
Girls	1.11 (0.60–2.05)	0.727	1.01 (0.53–1.93)	0.969
SOHI (plaque)	2.47 (0.97–6.27)	0.057	2.66 (1.01–7.02)	0.048
SOHI (calculus)	1.46 (.45–4.68)	0.524	0.85 (0.23–3.06)	0.805
	Goodness of fit test Hosmer–Lemeshow chi2(8) = 6.95, *p* = 0.5419			

* Reference category. OR = Odds Ratio. CI = Confidence Intervals.

## Data Availability

The data sets generated and analyzed during the current study are available from the corresponding author on reasonable request.

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
