# Peer review of "Experience and Prevalence of Dental Caries in 6 to 12-Year-Old School Children in an Agricultural Community: A Cross-Sectional Study"

_children, 2021, doi:10.3390/children8020099_

Round 1

Reviewer 1 Report

Introduction

  • I think you have detailed the caries problem in the cohort and justified your study.  I wouldn't change

Methods

  • I think you need to justify why 6 - 12 year olds? 
  • How can you be assured that all the children were from agricultural workers? I don't think you explain this in enough detail
  • Nice to see a sample size calculation - can you reference the pilot study?
  • What variables were collected on the questionnaire? Was the piloted prior to dissemination?  
  • Statistical analysis is described nicely, and seems appropriate. 
  • Independent variables (age 6 - 12 years), I am assuming this wasn't dichotomous like the other independent variables?  What about other variables that could explain differences? Number of children in household? Geographic access to a dental clinic (especially if some are very rural)

Results

  • I think the tables are quite confusing; there are lots of additional information in the legends that make it hard to understand. what is significant, what is not? what test has been done etc.
  • Is the use of mean and SD appropriate for ordinal data, which dmft and DMFT are?
  • There is no need to duplicate results in the text and tables e.g pg 4 line 155 is repeated in table 4

Discussion

  • This is ok - explains your results.  I do think you could look at increasing the evidence about dental health and access in marginalised groups. 
  • It might be worth discussing how alternative regions in Mexico might be different
  • Consider discussing alternative methodological approaches e.g. larger scale epidemiological study, or a study including non-agricultural 

Author Response

Reviewer 1

  1. I think you have detailed the caries problem in the cohort and justified your study.  I wouldn't change

Response: We are grateful for this observation.

Methods

  1. I think you need to justify why 6 - 12 year olds? 

Response: This is an age group commonly used for dental caries epidemiological research, in part because both dentitions are included. Being school age children, implications for preventive, health promotion, and restorative clinical programs become salient for policy decisions.

  1. How can you be assured that all the children were from agricultural workers? I don't think you explain this in enough detail

Response: In the new manuscript we now explain that we used a survey to ascertain children status.

  1. Nice to see a sample size calculation - can you reference the pilot study?

Response: Unfortunately the pilot study has not been published. It was undertaken on 70 children – 35 migrant, 35 non-migrant. Its goal was to establish prevalence figures to estimate values for sample size calculations.

  1. What variables were collected on the questionnaire? Was the piloted prior to dissemination?  

Response: The survey only contemplated age and sex. Other clinical variables were separately obtained, such as oral hygiene, fluorosis, malocclusion, and Community Periodontal Index (not reported in the present manuscript). Our regression models only included age, sex, migrant status, and oral hygiene.

  1. Statistical analysis is described nicely, and seems appropriate. 

Response: We are grateful for this observation.

  1. Independent variables (age 6 - 12 years), I am assuming this wasn't dichotomous like the other independent variables?  What about other variables that could explain differences? Number of children in household? Geographic access to a dental clinic (especially if some are very rural)

Response: We did not explore other independent variables. Recommendations from the reviewer have led us to start planning a subsequent study in which we will incorporate a more comprehensive battery of questions.

Results

  1. I think the tables are quite confusing; there are lots of additional information in the legends that make it hard to understand. what is significant, what is not? what test has been done etc.

Response: In the new manuscript we have simplified and complemented tables to make them easier to follow and more intuitive. We have put some of the relevant explanations in the text in order to not over saturate table content.

  1. Is the use of mean and SD appropriate for ordinal data, which dmft and DMFT are?

Response: Scale for both dmft and DMFT is quantitative discrete: values without decimals, which can be appropriately represented through mean and standard deviation. Scales range from 0 to 20 (dmft) and 0 to 28 (DMFT, without including third molars).

  1. There is no need to duplicate results in the text and tables e.g pg 4 line 155 is repeated in table 4

Response: In the new manuscript we have simplified and complemented tables to make them easier to follow and more intuitive – together with careful avoidance of duplication in text.

Discussion

  1. This is ok - explains your results.  I do think you could look at increasing the evidence about dental health and access in marginalised groups. 

Response: In the new manuscript we comply with this recommendation/observation.

  1. It might be worth discussing how alternative regions in Mexico might be different

Response: In the new manuscript we comply with this recommendation/observation.

  1. Consider discussing alternative methodological approaches e.g. larger scale epidemiological study, or a study including non-agricultural 

Response: In the new manuscript we comply with this recommendation/observation.

Reviewer 2 Report

The authors did a good job overall.

Although the background data they presented is accurate , in developed countries there has been a decrease in caries rates.
The paper may benefit from a better explanation of the caries experiences in children in Mexico first and then make the case of the uniqueness of the studied population and compare the characteristics of non migrant with the "city" children.

Author Response

Reviewer 2

  1. The authors did a good job overall. Although the background data they presented is accurate, in developed countries there has been a decrease in caries rates. The paper may benefit from a better explanation of the caries experiences in children in Mexico first and then make the case of the uniqueness of the studied population and compare the characteristics of non migrant with the "city" children.

Response: We are grateful for this observation. In the new manuscript we comply with this recommendation/observation. We would also like to note that while it is true that mean caries rates have generally decreased in developed countries, the entire phenomenon may be better represented by stating that caries experience has concentrated in smaller subsets of the population, therefore increasing health disparities overall.

Reviewer 3 Report

Dear authors, 

you paper entitled "Experience and prevalence of dental caries in 6—to 12-year-old  school children in an agricultural community: a cross-sectional study" is well written both in language and content. 

Introduction well reported the "state of the art"; however I suggest to report the impact of oral health-related quality of life on children, and the efeect of caries on their daily life and parental perception. for the purpose, I suggest the following references

Contaldo M, Della Vella F, Raimondo E, Minervini G, Buljubasic M, Ogodescu A, Sinescu C, Serpico R. Early Childhood Oral Health Impact Scale (ECOHIS): Literature review and Italian validation. Int J Dent Hyg. 2020 Nov;18(4):396-402. doi: 10.1111/idh.12451. Epub 2020 Jul 12. PMID: 32594620.

Chaffee BW, Rodrigues PH, Kramer PF, Vítolo MR, Feldens CA. Oral health-related quality-of-life scores differ by socioeconomic status and caries experience. Community Dent Oral Epidemiol. 2017 Jun;45(3):216-224. doi: 10.1111/cdoe.12279. Epub 2017 Jan 12. PMID: 28083880; PMCID: PMC5506781.

M&M section is well described and offers sufficient details for reproducing similar studies . However, I have a question:

  1. authors wrote "The sampling frame was made up of six public schools". could you, please, define/explain  how did you choose the six schools? were they randomly selected or based on other criteria? 

"results" are exhaustive, detailed and tables helped to summarize the main findings and differences between groups. 

"discussion" is the part I preferred: it is not only well written, but also criticized the limits of the study. I really appreciated this part. 

Author Response

Reviewer 3

  1. You paper entitled "Experience and prevalence of dental caries in 6—to 12-year-old  school children in an agricultural community: a cross-sectional study" is well written both in language and content. Introduction well reported the "state of the art"; however I suggest to report the impact of oral health-related quality of life on children, and the efeect of caries on their daily life and parental perception. for the purpose, I suggest the following references:  

Contaldo M, Della Vella F, Raimondo E, Minervini G, Buljubasic M, Ogodescu A, Sinescu C, Serpico R. Early Childhood Oral Health Impact Scale (ECOHIS): Literature review and Italian validation. Int J Dent Hyg. 2020 Nov;18(4):396-402. doi: 10.1111/idh.12451. Epub 2020 Jul 12. PMID: 32594620.

Chaffee BW, Rodrigues PH, Kramer PF, Vítolo MR, Feldens CA. Oral health-related quality-of-life scores differ by socioeconomic status and caries experience. Community Dent Oral Epidemiol. 2017 Jun;45(3):216-224. doi: 10.1111/cdoe.12279. Epub 2017 Jan 12. PMID: 28083880; PMCID: PMC5506781.

 Response: In the new manuscript we comply with this recommendation/observation, together with adding proposed citations. 

  1. M&M section is well described and offers sufficient details for reproducing similar studies. However, I have a question: authors wrote "The sampling frame was made up of six public schools". Could you, please, define/explain how did you choose the six schools? were they randomly selected or based on other criteria? 

Response: In the community targeted for the study there are six elementary schools – and all were included. No random selection was used.

  1. "results" are exhaustive, detailed and tables helped to summarize the main findings and differences between groups. 

Response: We are grateful for this observation.

  1. "discussion" is the part I preferred: it is not only well written, but also criticized the limits of the study. I really appreciated this part. 

Response: We are grateful for this observation.

Reviewer 4 Report

  1. Abstract:

Based on which of the results did the authors conclude that the oral health status of children from agricultural manual worker households was poor? The conclusions should be based on the results of this study and should present an answer to the research question of this study.

  1. Introduction:

Lines 46-48: It should be mentioned to what region of the world this statement applies, in accordance with the content of References 3 and 4.

  1. Materials and Methods:

The clinical examination was performed under natural light. Are there any references to prove that natural light would be enough to detect all the caries? If no, could the authors estimate the proportion of caries that might have been overlooked because of insufficient light?

  1. Discussion:

1) Lines 180-181: What do the authors mean by “caries prevalence was considerable in our study population”? Considerable compared to what other population and at what timepoint (at present or in the past?) Relevant data from the literature should also be provided to support this statement. (The references provided in the next lines, references 3 and 18, rather suggest that the caries level found in this study is at the lower end of average dmft and DMFT indexes for similar age groups in Mexico.)

2) Lines 189-192, lines 196-197, and lines 198-210: These statements support the need for this study rather than discussing its results, so they would better fit into the Introduction section.

3) Lines 212-214: The subjects all had the same social background, so how could the social structure moderate the influence of migration? Also, the statement that access to health services could moderate the influence of migration comes in contradiction with the other statements of the study and the cited literature. Or, do the authors mean that the program mentioned in lines 214-216 could bridge the gaps in access to oral health between the two studied populations? If so, clear arguments should be provided.

The possibility of differences in sugar ingestion and effect of fluoride (from toothpastes or drinking water, if relevant) between the two study populations should be also discussed.

  1. Conclusions:

Same comment as for the Abstract and the first comment of the Discussion.

Author Response

Reviewer 4

Abstract:

  1. Based on which of the results did the authors conclude that the oral health status of children from agricultural manual worker households was poor? The conclusions should be based on the results of this study and should present an answer to the research question of this study.

Response: This conclusion was based on the fact that more than 50% of children had caries experience but little restorative treatment for existing caries. Both on account of preventing carious lesions and having access to dental treatment to address existing disease, the subjective appraisal of poor oral health status appears justified. In the new manuscript we present a summative statistic for both caries indices.

Introduction:

  1. Lines 46-48: It should be mentioned to what region of the world this statement applies, in accordance with the content of References 3 and 4.

Response: In the new manuscript we comply with this recommendation/observation.

Materials and Methods:

  1. The clinical examination was performed under natural light. Are there any references to prove that natural light would be enough to detect all the caries? If no, could the authors estimate the proportion of caries that might have been overlooked because of insufficient light?

Response: We followed the entries in the WHO “Oral Health Surveys Basic Methods”, namely:

“The lighting should be as consistent as possible throughout the survey. If electricity is available at all locations, a lightweight portable examination light (in the blue-white colour spectrum) should be used. Inflammatory and structural changes of the oral tissues are more difficult to detect under normal artificial light (yellow-red in colour) than under natural or corrected artificial light. If electricity or battery-operated lights are not available at some survey sites, natural light should be used at all locations.”

The reference has been added

Discussion:

  1. 1) Lines 180-181: What do the authors mean by “caries prevalence was considerable in our study population”? Considerable compared to what other population and at what timepoint (at present or in the past?) Relevant data from the literature should also be provided to support this statement. (The references provided in the next lines, references 3 and 18, rather suggest that the caries level found in this study is at the lower end of average dmft and DMFT indexes for similar age groups in Mexico.)

Response: We have stated that in the absence of prior research incorporating similar study groups, we resorted to the general population. In the new manuscript we indicate that caries indices and prevalence observed were within ranges reported previously for primary teeth – but not for permanent teeth; the latter were lower than those observed in the general Mexican population.

  1. 2) Lines 189-192, lines 196-197, and lines 198-210: These statements support the need for this study rather than discussing its results, so they would better fit into the Introduction section.

Response: In the new manuscript we comply with this recommendation/observation.

  1. 3) Lines 212-214: The subjects all had the same social background, so how could the social structure moderate the influence of migration? Also, the statement that access to health services could moderate the influence of migration comes in contradiction with the other statements of the study and the cited literature. Or, do the authors mean that the program mentioned in lines 214-216 could bridge the gaps in access to oral health between the two studied populations? If so, clear arguments should be provided.

Response: Overall, all participating subjects had similar socioeconomic positions. The main differentiating factor is that families who have migrated to the study location originated from some of the poorest areas in Mexico. An interesting findings is that despite such fundamental difference, caries experience was not substantially different between the two groups. While public assistance program may have attenuated differentials, it is possible that after a few years of residence in a different area, structural factors level the playing field for dental caries across groups. We have attempted to clarify these perspectives in the new manuscript.

  1. The possibility of differences in sugar ingestion and effect of fluoride (from toothpastes or drinking water, if relevant) between the two study populations should be also discussed.

Response: In the new manuscript we comply with this recommendation/observation.

Conclusions:

  1. Same comment as for the Abstract and the first comment of the Discussion.

Response: This conclusion was based on the fact that more than 50% of children had caries experience but little restorative treatment for existing caries. Both on account of preventing carious lesions and having access to dental treatment to address existing disease, the subjective appraisal of poor oral health status appears justified. In the new manuscript we present a summative statistic for both caries indices.
